# Engineered Expression of Vip3A in Green Tissues as a Feasible Approach for the Control of Insect Pests in Maize

**DOI:** 10.3390/insects14100803

**Published:** 2023-10-06

**Authors:** Guangsheng Yuan, Cheng Zeng, Haoya Shi, Yong Yang, Juan Du, Chaoying Zou, Langlang Ma, Guangtang Pan, Yaou Shen

**Affiliations:** State Key Laboratory of Crop Gene Exploration and Utilization in Southwest China, Key Laboratory of Biology and Genetic Improvement of Maize in Southwest Region of Ministry of Agriculture, Maize Research Institute, Sichuan Agricultural University, Chengdu 611130, China

**Keywords:** genetically modified, Cre/*lox*P, site-specific excision, transgenic maize, insect-resistance

## Abstract

**Simple Summary:**

The use of genetically modified (GM) crops expressing *Bacillus thuringiensis* (Bt) proteins is currently the most common method for controlling herbivorous pests. However, the insecticidal proteins in the edible tissues of transgenic crops has become an issue of intense public concern. We developed an available strategy based on the Cre/*lox*P-mediated system for manipulating *Vip3A* expression in maize defined green tissues outside edible parts to relieve public concerns. To test this approach, two basic transgenic maize named KEY and LOCK containing Cre and Vip3A, were generated, respectively. By crossing the KEY and LOCK plants, the expression of blocked Vip3A was enabled in specific green tissues in their hybrids. After assessing the insect-resistance of the transgenic maize in the laboratory and field, the KEY × LOCK hybrids showed high plant resistance levels against the two common lepidopteran pests. The present study suggested that the Cre/*lox*P-mediated genetic engineering approach would be crucial in ensuring the biosafety of GM plants, and also provided an effective strategy for manipulating transgene expression in specific tissues.

**Abstract:**

Genetic engineering technology offers opportunities to improve many important agronomic traits in crops, including insect-resistance. However, genetically modified (GM) exogenous proteins in edible tissues of transgenic crops has become an issue of intense public concern. To advance the application of GM techniques in maize, a Cre/*lox*P-based strategy was developed for manipulating the transgenes in green tissues while locking them in non-green tissues. In the strategy, the site-specific excision can be used to switch on or off the expression of transgenes at specific tissues. In this work, two basic transgenic maize, named KEY, carrying the *Cre* gene, and LOCK, containing the *Vip3A* gene with a blocked element, were obtained based on their separate fusion gene cassettes. The expression level and concentration of Vip3A were observed with a high specific accumulation in the green tissues (leaf and stem), and only a small amount was observed in the root and kernel tissues in the KEY × LOCK hybrids. The insect resistance of transgenic maize against two common lepidopteran pests, *Ostrinia furnacalis* and *Spodoptera frugiperda*, was assessed in the laboratory and field. The results indicate that the hybrids possessed high resistance levels against the two pests, with mortality rates above 73.6% and damage scales below 2.4 compared with the control group. Our results suggest that the Cre/*lox*P-mediated genetic engineering approach has a competitive advantage in GM maize. Overall, the findings from this study are significant for providing a feasible strategy for transgenes avoiding expression in edible parts and exploring novel techniques toward the biosafety of GM plants.

## 1. Introduction

Maize is a primary staple food crop that plays a pivotal role in global food security. In regions where maize is grown, various biotic stresses can impact both production capacity and safety, including the corn borer (*Ostrinia furnacalis* Guenée (Crambidae)) and the fall armyworm (*Spodoptera frugiperda* (J. E. Smith)), resulting in yield and quality losses [1,2,3]. Currently, general management for preventing the threat of pests relies primarily on chemical insecticides, which may lead to possible environmental and health hazards [4]. The use of genetically modified (GM) crops expressing *Bacillus thuringiensis* (Bt) proteins is the most common method for controlling herbivorous pests [5,6,7]. Transgenic maize expressing the Vip3A (vegetative insecticidal protein 3) protein from Bt is widely employed in GM crops for controlling lepidopteran pests and has been planted for more than two decades [8,9,10]. As a result of their high effectiveness in pest control, transgenic Bt crops have seen consistent growth since their initial commercialization in the USA during the 1990s [11]. While Bt maize has been extensively cultivated in numerous countries in recent decades, its biosafety remains a subject of controversy in certain regions [12,13]. One possible reason is that exogenous proteins of GM crops are expressed simultaneously in various tissues, including the edible parts, which would lead to potential risks to the environment or consumer health [14].

To relieve public concerns about GM crops, several genetic engineering techniques have been developed to manipulate transgene expression in confined tissues [15,16,17,18,19,20,21,22]. For instance, the Cre/*lox*P site-specific recombination system is the most-mentioned tool for precisely manipulating DNA introduced into transgenic plants due to its simplicity and efficiency [23,24]. In the system, Cre recombinase, responsible for specific excision of the sequences embedded by directly repeated *lox*P sites, has been applied to remove transgenes from the host genome [25]. The transcription terminator named *Agrobacterium* nopaline synthase terminator (*NosT*) is the most common blocked element in plant genetic engineering that can efficiently halt the transcription process of a gene [26]. Based on the system, a new strategy has been developed to enable the expression of transgenes in desired tissues, depending on the site-specific excision reaction occurring at particular places. Here, two basic transgenic plants, individual Cre and *lox*P, are required for performing a site-specific excision reaction in their heterologous system [27,28,29]. The *lox*P plants that carry transgenes blocked by a *NosT* terminator can be crossed with Cre plants that express Cre activity, in which case the terminator can then be excised away in their F_1_ progeny. In this case, the blocked transgenes in the *lox*P plants can be activated and allowed to express themselves after the excision reaction at specific places. Therefore, the presence of Cre in particular tissues plays a key role in determining the tissue-specific expression of transgenes. The *Cre* gene can be driven by diverse promoters and then expressed in specific tissues, and could prompt the excision reaction occurring in defined tissues, thereby enabling the expression of blocked transgenes in the target tissues. The promising Cre/*lox*P-mediated system has previously been employed in transgenic tobacco, rice, and oilseed rape to regulate the expression of target genes in specific tissues [27,28,30].

Here, we developed a comprehensive strategy based on the Cre/*lox*P-mediated system for manipulating transgene expression in maize green tissues outside the edible parts to relieve public concerns. In this work, the transgene *Vip3A* was explicitly expressed in the green tissues owing to the presence of Cre in the specific tissues when the two basic transgenic maize were brought together. Moreover, the crossed hybrids were evaluated with high resistance against insect pests. These findings illustrate a feasible strategy for manipulating the expression of transgenes in specific tissues and also provide a valuable tool for GM safety in other transgenic plants.

## 2. Materials and Methods

### 2.1. Construction of the Fusion Gene Cassette

The green-tissue-specific promoter *Zm1rbcS* was chosen according to previous relevant studies [27,31], given its excellent characteristics in driving specific expression in green tissues of the related genes. Under the control of the *Zm1rbcS* promoter, a cassette named pKEY was constructed to regulate *Cre* gene expression in green tissues. Another cassette named pLOCK, containing *NosT* inserted between the pairwise *lox* sites following the transgene Vip3A, was driven by the strong constitutive promoter ZmUbi (Figure 1). Herein, the sequences of genes *Cre* and *Vip3A* were kindly provided by Dr. Chen and Prof. Tu (Zhejiang University, China). The two fusion cassettes were created in plant vector pCAMBIA3300 and then transferred into *Agrobacterium tumefaciens* strain EHA105 [32]. The neomycin phosphotransferase gene (*nptII*) was a marker gene for the selection of transgenic plants during transformation. The vectors and primers are listed in Appendix A.

### 2.2. Plant Material and Transformation

Each constructed cassette was introduced into immature embryos of the wild-type maize inbred line KN5585. The individual transgenic maize containing *Cre* and *Vip3A* were called KEY and LOCK plants. Transgenic plants were either self-pollinated or crossed to generate KEY × LOCK hybrids. All of the plants were grown in the transgenic base of Sichuan Agricultural University (Chengdu, Sichuan Province, China) (30°43′11.06″ N; 103°52′12.30″ E). Each of the plants were sown in single 3 m row plots, with a distance of 50 cm between plants within a row and 75 cm between rows, following a randomized block design. No pesticides were applied during the maize growth period.

### 2.3. Tissue Expression in Transgenic Plants

The expression levels of *Cre* and *Vip3A* were detected in various tissues (root, stem, leaf, sheath, husk, silk, and kernel) of the different transgenic maize. The total RNAs from different maize tissues were extracted using the TRIzol reagent (Invitrogen, Waltham, MA, USA) according to the manual. A quantitative real-time PCR (qRT-PCR) was performed, with 18S rRNA as the internal control. The thermal cycle conditions were as follows: 2 min at 95 °C followed by 40 cycles of 15 s at 95 °C, 15 s at 56–57 °C, and 15 s at 72 °C. Each PCR reaction was repeated with three technical replicates.

### 2.4. Concentration of the Vip3A Protein

The concentration of the Vip3A protein in the various tissues (root, stem, leaf, sheath, husk, silk, and kernel) of the different transgenic maize was determined using the Enzyme-linked immunosorbent assay (ELISA) (Agdia^®^, Elkhart, IN, USA). Each sample was added to a 96-well ELISA microplate in triplicate. The protein concentration was calculated using a linear regression equation for the standard curve including only the triplicates with a coefficient of variation lower than 20%.

### 2.5. Bioassay in Laboratory

We performed a diet-overlay bioassay to evaluate the insect resistance of transgenic maize against two common lepidopteran pests—*Ostrinia furnacalis* and *Spodoptera frugiperda*. In Southwest China, transgenic maize is not planted. The original populations of the two pests were collected from this area and they had not previously been exposed to Bt proteins. The two pests were reared on an artificial diet in the laboratory and maintained under controlled conditions (27 ± 1 °C, 14: 10 light/dark photoperiod, and 70–75% RH). Eggs from the two pests were commercially acquired (Keyun Biotechnology Co., Ltd., Jiyuan, China) and reared in the laboratory. When the plants reached the V6 growth stage, bioassays were undertaken to evaluate the resistant response of transgenic maize. In the laboratory, fresh leaves of different maize were cut into pieces of approximately 3 cm^2^ and placed into a 24-well bioassay tray (Tsingke Biotechnology Co., Ltd., Beijing, China). One first-instar larva of a single pest was added to each well of the tray for every maize plant. Three experimental plant leaves from the KEY × LOCK hybrids and controls were considered replicates, and three replicates for each maize were designated for a single larval infestation in the insect bioassay. The non-transgenic maize KN5585 and individual LOCK plants served as the negative and positive controls, respectively. The number of living larvae, dead larvae, and their larval stages was observed and recorded after infestation. If a larva had not developed beyond the first instar, it would be counted as dead. Larval mortality was calculated as mortality (%) = 100 × number of dead larvae plus the number of surviving larvae still in the first instar divided by the total number of pests assayed [9]. The mortality rate was corrected for each kind of maize using Abbott’s formula [33].

### 2.6. Field Tests for Insect Resistance

We also performed a field test in addition to characterizing the resistance of transgenic maize. The same two pests of *O. furnacalis* and *S. frugiperda* were inoculated into each maize when the plants reached the V_6_ stage under field conditions. For each maize, including the KEY × LOCK hybrids and controls, 30 plants were chosen randomly for a single pest implementation as a replicate, and three replicates were conducted for a single larval infestation in the field test. Each plant was infested with 60 first-instar larvae on the whorls, and the plant leaf damage scale of each maize was calculated as the mean score of the inoculated maize. The KN5585 and LOCK plants were the controls. The damage index of the plants was surveyed 14 days after artificial infestation and calculated using a 0–9 whole-plant leaf damage scale according to the described study [34].

### 2.7. Statistical Analyses

The data for the Vip3A protein concentration were compared with the control using Student’s *t*-test. The percentages of larval mortality and plant leaf damage scale were analyzed using a one-way analysis of variance (ANOVA) model to determine the insect-resistant response variable between the hybrid and controls. The data of insect-resistance in each maize were conducted using the Dunnett multiple comparison method for a significant difference test (*p* < 0.05). All of the statistical analyses were conducted using SPSS 21.0 (http://www.spss.com) (accessed on 23 July 2023).

## 3. Results

### 3.1. Fusion Gene Cassette and Plant Transformation

The two constructed gene cassettes were each built into the plant vector pCAMBIA3300 (Appendix A). The green-tissue-specific promoter *Zm1rbcS* in the pKEY cassette was used to drive the Cre recombinase expression specifically in green tissues (Figure 1). The other pLOCK cassette comprised the strong constitutive promoter *ZmUbi* for driving the exogenous *Vip3A* gene, in which *NosT* was inserted between the pairwise *lox* sites to block the expression of the transgene (Figure 1). Each fusion cassette consisted of transformed immature embryos of the maize inbred line KN5585 background. Putative transgenic plants were initially screened for kanamycin resistance and then confirmed with molecular characterization. A total of seven individual events for pKEY and 140 events for pLOCK were produced, respectively. PCR analyses of the transgenic events showed that the expected amplicons corresponding to genes *Cre* and *Vip3A* were present in their transformed lines (Appendix A). The transgenic maize containing *Cre* and *Vip3A* were called KEY and LOCK plants, respectively. Finally, these transgenic plants were subsequently self-pollinated to obtain advanced generations.

**Figure 1 insects-14-00803-f001:**
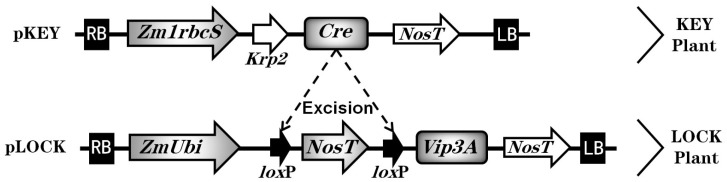
Construction of gene cassettes and schematic diagram of the Cre/*lox*P-mediated strategy. In the pKEY cassette, the Cre gene was driven by the green-tissue-specific promoter Zm1rbcS and expressed in specific green tissues under the control of Zm1rbcS. In the other pLOCK cassette, the exogenous Vip3A gene was blocked by an inserted NosT between the pairwise *lox* sites, driven by the strong constitutive promoter ZmUbi. NosT as a blocked element was introduced in front of the Vip3A gene to keep the transgene at rest. The two fused cassettes were each transformed into the wild-type inbred line KN5585 to generate individual transgenic maize, called KEY and LOCK plants, respectively. By crossing the KEY and LOCK plants, the expressed Cre recombinase could lead to the deletion of *lox*P-NosT; thus, a site-specific excision reaction could occur in specific green tissues in their F_1_ progeny. NosT should then be excised away, thereby enabling the expression of the blocked Vip3A in the defined tissues.

### 3.2. Tissue-Specific Expression of Transgenes

The expression levels of *Cre* and *Vip3A* in various tissues (root, stem, leaf, sheath, husk, silk, and kernel) from advanced progenies of the individual KEY or LOCK plants and their KEY × LOCK hybrids were checked using qRT-PCR. Higher expression levels of *Cre* appeared in the leaf, followed by the stem and sheath, and a small amount of expression was found in the root, silk, and kernel, indicating that the *Cre* gene under the control of the green-tissue-specific promoter *Zm1rbcS* was activated specifically in the green tissues in KEY plants and KEY × LOCK hybrids (Figure 2a). On the other hand, the expression of *Vip3A* was observed at a low level in various tissues of LOCK plants, which might be explained by the fact that the transgenes were inhibited by the inserted *NosT* (Figure 2b). However, the expression of *Vip3A* in the KEY × LOCK hybrids was detected at a high level in the leaf and stem. Still, the rare accumulation in other tissues, including the root and kernel, reflects that the expression level of *Vip3A* in the green tissues of the hybrids is extensively activated compared with those in individual LOCK plants.

### 3.3. The Concentration of the Insecticidal Protein

We performed the ELISA method to track insecticidal protein in the various tissues (root, stem, leaf, sheath, husk, silk, and kernel) of the transgenic maize. In the KEY × LOCK hybrids, the mean concentration of the Vip3A protein reached 9.59 ng/g, with 6.13 ng/g in the leaf and stem, and only 0.68 ng/g and 0.71 ng/g in the root and kernel, respectively (Table 1). Furthermore, the protein was detected as being maintained at a low level (ranging from 0.43 ng/g to 0.74 ng/g) in the LOCK plant tissues. Expectedly, the content pattern of the Vip3A protein in the hybrids was similar to the model of the above tissue expression. From these results, the leaf and stem tissues contained higher protein levels than those in the root and kernel parts, reflecting that the Vip3A protein has a specific accumulation in green tissues. After comparing the concentration of Vip3A protein in the different trangenic maize, the Vip3A protein was probably blocked at a low expression in the LOCK plants, whereas it might have been activated in the leaf and stem tissues when the KEY and LOCK plants were brought together.

### 3.4. The Efficiency of Insect Resistance in Transgenic Maize

A bioactivity test was conducted using the larvae of two major insect pests *O. furnacalis* and *S. frugiperda* to examine insect-resistant efficiency of the transgenic maize. The neonate larvae (first instar) of the two insect pests were single manually fed in the laboratory with fresh leaves of the hybrids and control maize, and the mortality rate was evaluated after infestation via the diet-overlay bioassay. The leaf damage of the maize in the control trays became apparent after two days and was heavily damaged after three days of implementation (Figure 3a,b,d,e). In detail, the mean mortality rates of the *O. furnacalis* larvae fed with the control plants were 8.3% and 10.6% (Table 2). Similarly, the mortality rates of *S. frugiperda* were 8.7% and 9.2% in the control trays, respectively (Table 2). No significant effects of the mean mortality rates were detected in the control maize. At the same time, the mean mortality of the *O. furnacalis* and *S. frugiperda* larvae fed on the KEY × LOCK hybrids was overall high, with rates of 81.2% and 73.6% after three days of implementation, respectively (Figure 3c,f, Table 2). Statistical analyses showed a significant difference in mean mortality rates between the hybrids and control maize (*O. furnacalis*: F_2,6_ = 349.61, *p* < 0.001; *S. frugiperda*: F_2,6_ = 278.69, *p* < 0.001) (Table 2). The results showed that the hybrids exhibited a highly resistant performance regarding the protection of maize with almost no damage and that they were significantly better than those of the controls.

To further confirm the insect resistance response of the transgenic maize, the larvae of the same two pests, *O. furnacalis* and *S. frugiperda*, were used in the field. Obvious damage was observed in the control plants after 14 days of implementation when the two instar larvae were introduced onto the maize whorls (Figure 4a,b,d,e). Damaged scales from *O. furnacalis* infestation were 7.5 (KN5585) and 7.7 (LOCK), respectively, and there were no significant differences among them (Table 2). Similarly, the damaged scales caused by *S. frugiperda* showed no significant differences among the controls with 8.1 and 7.9, respectively (Table 2). In contrast, the damaged scales in the hybrids were only 2.4 for *O. furnacalis* and 1.6 for *S. frugiperda*, respectively, suggesting that the hybrids could increase the protective efficacy of the maize (Figure 4c,f). Statistical analyses showed a significant difference in the plant leaf damage scale between the hybrids and control maize (*O. furnacalis*: F_2,6_ = 117.58, *p* < 0.001. *S. frugiperda*: F_2,6_ = 196.43, *p* < 0.001) (Table 2). The insect-resistant performance in the field followed the results of the diet-overlay bioassay in the laboratory. High mortality rates and low damage scales were recorded during the infestation, implying that the hybrids may have a sufficient insecticidal Vip3A protein to protect the maize against the two lepidopteran pests.

## 4. Discussion

As GM foods are becoming present in our diets, their biosafety has attracted significant public concern [23,35]. We focused on the insect-resistant transgenic maize as a lack of transgenes in the edible parts may be more acceptable to the public. The Cre/*lox*P-mediated system provides an available strategy for manipulating the expression of transgenes in defined green tissues outside of the kernel. For this approach, two separate gene cassettes were first constructed to generate two basic transgenic maize plants—KEY and LOCK (Figure 1). By crossing them, the excision reaction should result in the locked transgenes being opened in specific tissues. The comprehensive design here also appeared in previous studies. For instance, Luo et al. successfully constructed the Cre/*lox*P with a pollen and seed-specific promoter system to remove functional transgenes from the target tissues [29]. Another study used the Cre/*lox*P and seed-specific cruciferin C promoter resulting in 10% marker-free transgenic tobacco plants [30]. Chen et al. developed a Cre/*lox*P gene switch system to limit Cry protein in the green tissues of rice [27]. Boszorádová et al. combined the Cre/*lox*P and the embryo-specific *CRUC* promoter to remove hazardous transgenes from the genome in commercial oilseed rape [28]. For this comprehensive strategy, diverse promoters are crucial regulatory elements and have an efficient impact on the expression of transgenes in defined tissues [36,37,38]. In the current study, the green-tissue-specific promoter *Zm1rbcS* with a moderate drive capability was chosen to induce the expression of the simple gene *Cre* in the KEY plants. However, the exogenous gene *Vip3A* required a powerful promoter to produce a sufficient quantity of insecticidal protein. Therefore, by taking advantage of the characteristics of the two types of promoters, the problem of tissue-specific expression compatible with insect resistance for GM maize can be resolved through the pyramiding of KEY and LOCK plants.

The presence of Cre protein in green tissues is necessary for the effective control of Vip3A protein in defined tissues. In our study, the *Cre* gene was expressed specifically in green tissues, including the leaf and stem, with low levels in the root and kernel, suggesting that the gene should be specifically induced to express at green tissues under the control of *Zm1rbcS* (Figure 2a). Moreover, the expression levels of *Vip3A* were observed at a low level in the various tissues of LOCK plants, but high levels in the KEY × LOCK hybrids leaf and stem. These results reflected that the *Vip3A* might be activated in target green tissues and locked in non-green tissues of the hybrids (Figure 2b). Meanwhile, the concentration of Vip3A protein is largely correlated with the Cre expression levels in the hybrids (Table 1). It was speculated that the excision reaction might have occurred in target green tissues when the KEY and LOCK plants were brought together. Then, the inserted *NosT* in LOCK plants might be removed due to Cre presence, thereby allowing expression of the blocked *Vip3A* in the excision tissues. It should be noted that the green-tissue-specific promoter does not completely control the accumulation of *Vip3A* due to its background expression in non-green tissues, thus leading to a trace amount in root and kernel tissues. These results are parallel to other similar studies in transgenic plants. For instance, the amount of GM protein detected in *rbcS*-driven *Cry1C* rice endosperm was 2.6 ng/g in the tested samples [39], and *PEPC*-driven *Cry1Ab* maize kernels were 15–18 ng/mg soluble protein [40]. A trace amount of transgene product was also observed in *PNZIP*-driven Bt-cotton seeds and *PDX1*-driven GUS-rice endosperm [41,42]. Our Cre/*lox*P-mediated approach has significantly minimized the presence of exogenous Vip3A protein in edible parts, although it remains a slight imperfection. However, this was a preliminary step in manipulating *Vip3A* expression in maize green tissues. Further work including promoting the effectiveness of the green-tissue-specific promoter, blocked capacity of the *NosT*, and excision efficiency of Cre, is needed. In brief, these results offered an alternative strategy to manipulate transgenes expression at specific maize tissues outside of edible parts by combing the two individual transgenic plants.

Regarding the efficiency of the insecticidal protein based on the Cre/*lox*P-mediated approach, insect-resistance of the transgenic plants against pests attack was evaluated in the laboratory and field. High mortality rates and low damaged scales were recorded in the hybrids compared with the control group (Figure 3 and Figure 4). The results demonstrated that the hybrids showed a high degree of resistance against the two lepidopteran pests, implying that the Cre/*lox*P-mediated approach was effective for generating new transgenic maize with good resistance. These novel insect-resistant transgenic maize hybrids could serve as elite germplasm resources for extensive Bt application in GM maize breeding, as well as for the research of plant genetic engineering [41], control of gene expression [27], and so on. It should be noted that although there were significant differences in larval mortality and plant leaf damage scale between the hybrids and controls, the insect-resistant level of the hybrids was only moderate and likely would not reach commercial efficacy. The most likely reason was the lack of an adequate excision reaction between Cre and *NosT*, thus leading to the insecticidal Vip3A protein in the maize green tissues not reaching a high level. In the next step, a better promoter could be chosen to produce high level Cre for increasing the Vip3A protein in green tissues. On the other hand, the crossed hybrids with a high level of insect resistance could be selected for further breeding programs and potential commercial implementation.

## 5. Conclusions

In the expected development and commercial exploitation of insect-resistant transgenic maize, the “excess” exogenous protein should preferably be outside of the edible parts, which might be more acceptable to the public. Tissue-specific accumulation of the Vip3A protein in GM maize may help control the occurrence of pest attacks and limit the number of toxins in inedible tissues. Our results advocate that such a Cre/*lox*P-mediated approach can be recommended in plant genetic engineering. The strategy has several advantages through combining KEY and LOCK plants to manipulate hazardous transgene expression in the target tissues. Firstly, Cre recombinase expression in KEY plants is mainly accumulated in the defined tissues under strict control of the tissue-specific promoters, leading to subsequent excision in the targeted tissues. Second, the Cre/*lox*P-mediated system is usually applied to delete the introduced transgenes from the host genome, and the removal process may have potential risks by causing plenty of damage to the plants. Here, the transgene *Vip3A* is always present in the genome, with its expression selectively activated in the green tissues without removing DNA from genome. This has a minimal effect on host performance and could be applied more widely. Third, the tissue-specific promoter in KEY plants or blocked transgenes in LOCK plants can be replaced conveniently according to different demands for tissue-specific expression or desired traits. In these aspects, the current Cre/*lox*P-mediated approach could provide more options for manipulating the transgene expression in specific tissues and advance the plant genetic engineering concept closer to commercial implementation for GM maize.

## Figures and Tables

**Figure 2 insects-14-00803-f002:**
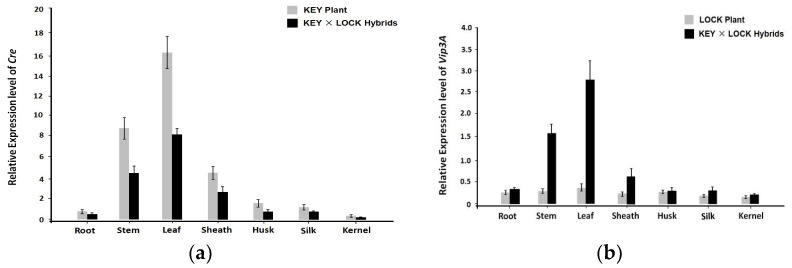
Expression pattern analyses of *Cre* and *Vip3A* genes in various tissues of different transgenic maize using qRT-PCR. (**a**) Tissue-specific expression profiles of *Cre* in multiple tissues (root, stem, leaf, sheath, husk, silk, and kernel) at the mature stage of the KEY × LOCK hybrids and KEY plants. (**b**) Expression profiles of *Vip3A* in various tissues of the KEY × LOCK hybrids and LOCK plants. Data are given as the mean ± SE (standard error) of three biological replicates.

**Figure 3 insects-14-00803-f003:**
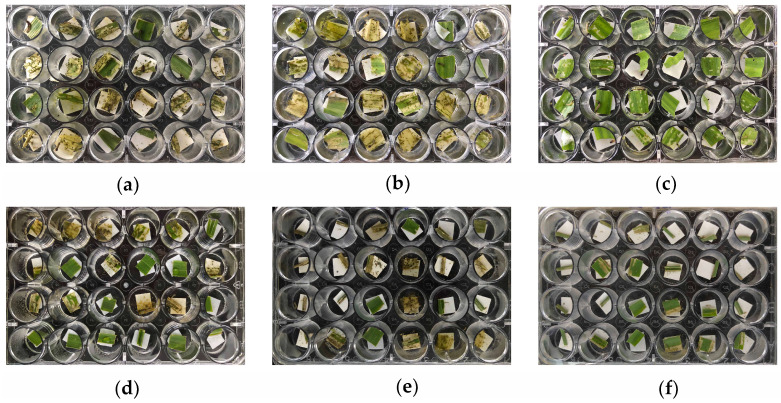
Insect resistance of different maize leaves against *O. furnacalis* and *S. frugiperda* using a diet-overlay bioassay in the laboratory. (**a**–**c**) Typical performance of different maize leaves damaged by the pest of *O. furnacalis* after three days of implementation. (**a**) Negative control KN5585. (**b**) Positive control LOCK. (**c**) Low damaged KEY × LOCK hybrids leaf. (**d**–**f**) Typical performance of different maize leaves damaged by another pest of *S. frugiperda* after three days of implementation. (**d**) KN5585. (**e**) LOCK. (**f**) Low damaged KEY × LOCK hybrids leaf.

**Figure 4 insects-14-00803-f004:**
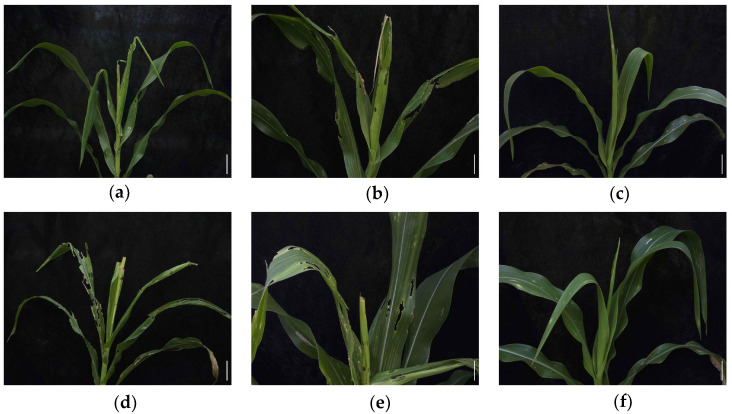
Field performance of different maize against *O. furnacalis* and *S. frugiperda* after manual infestation. (**a**–**c**) Insect-resistant performance of different maize against the pest *O. furnacalis* infestation. (**a**) Negative control: KN5585. (**b**) Positive control: LOCK. (**c**) Low damaged KEY × LOCK hybrids. (**d**–**f**) Insect-resistant performance of different maize against another pest, *S. frugiperda* infestation. (**d**) KN5585. (**e**) LOCK. (**f**) Low damaged KEY × LOCK hybrids. Scale bar = 10 cm.

**Table 1 insects-14-00803-t001:** The concentration of Vip3A protein in the LOCK plants and the KEY × LOCK hybrids ^a^.

Plant Material	Root (ng/g)	Stem (ng/g)	Leaf (ng/g)	Sheath (ng/g)	Husk (ng/g)	Silk (ng/g)	Kernel (ng/g)
LOCK	0.73 ± 0.12	0.51 ± 0.07	0.43 ± 0.09	0.74 ± 0.13	0.62 ± 0.07	0.68 ± 0.11	0.59 ± 0.08
KEY × LOCK	0.68 ± 0.23	6.13 ± 0.41 **	9.59 ± 0.38 **	1.08 ± 0.29	0.91 ± 0.35	0.74 ± 0.15	0.71 ± 0.12

^a^ Values are given as the mean ± SE (standard error). LOCK transgenic maize is a positive control. Multiple comparisons of the KEY × LOCK hybrids with the control maize using Student’s *t*-test. ** Means within a column significantly differed from LOCK transgenic maize at *p* < 0.01.

**Table 2 insects-14-00803-t002:** The mean mortality rate of *O. furnacalis* and *S. frugiperda* larvae via the diet-overlay bioassay in the laboratory and the plant leaf damage scale caused by the two pests in the field test ^a^.

Plant Material	Larval Mortality (%)		Plant Leaf Damage Scale	
	*O. furnacalis*	*S. frugiperda*	*O. furnacalis*	*S. frugiperda*
KN5585	8.3 ± 0.9 b	8.7 ± 0.7 b	7.5 ± 0.4 b	8.1 ± 0.5 b
LOCK	10.6 ± 1.4 b	9.2 ± 1.2 b	7.7 ± 0.5 b	7.9 ± 0.3 b
KEY × LOCK	81.2 ± 3.7 a	73.6 ± 2.9 a	2.4 ± 0.1 a	1.6 ± 0.1 a

^a^ Data are given as the mean ± SE (standard error). Wild-type KN5585 is the negative control, and individual LOCK transgenic maize is the positive control. Means within a column for each kind of maize followed by the different letters are significantly different according to one-way ANOVA and Dunnett comparison (*p* < 0.05). Each implementation included three replicates. For larval mortality, the means of the KEY × LOCK hybrids are significantly higher compared with the control maize group (*O. furnacalis*: F_2,6_ = 349.61, *p* < 0.001; *S. frugiperda*: F_2,6_ = 278.69, *p* < 0.001). For the plant leaf damage scale, the means of the hybrids are significantly lower compared with the control maize group (*O. furnacalis*: F_2,6_ = 117.58, *p* < 0.001. *S. frugiperda*: F_2,6_ = 196.43, *p* < 0.001).

## Data Availability

Data are available upon request.

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
