# Peer review of "Engineered Expression of Vip3A in Green Tissues as a Feasible Approach for the Control of Insect Pests in Maize"

_insects, 2023, doi:10.3390/insects14100803_

Round 1
Reviewer 1 Report
>>>The method described in the manuscript, which is based on the Cre/loxP-mediated system for manipulating gene expression in maize, is not inherently novel as the Cre/loxP system has been widely used in genetic engineering for many years. This system is a well-established and proven tool for controlling gene expression in various organisms, including plants. Therefore, the novelty of the research likely lies in the specific application of the Cre/loxP system to manipulate the expression of the Vip3A gene in maize, with a focus on confining its expression to specific tissues.
In terms of references, it is essential in scientific research to provide a solid foundation of relevant literature to support the research idea and demonstrate awareness of existing work in the field. Upon reviewing the manuscript, it appears that the authors have included references to key studies and relevant research. They reference the use of the Cre/loxP system in other transgenic plants like tobacco, rice, and oilseed rape, which helps establish the context for their research.
However, it's always a good practice to ensure that the references cover a broad spectrum of related work, especially if there have been significant developments in the field that may impact the current research. Additionally, the authors could further strengthen their argument by discussing how their approach builds upon or differs from previous applications of the Cre/loxP system in plants.The inclusion of relevant references supports the research idea, but authors should ensure that they cover the most up-to-date and pertinent literature in their field of study.
also, the provided excerpt from the manuscript mentions the assessment of pest control efficiency in hybrid maize but does not provide specific details regarding the experimental setup or the sample size. To evaluate the methodology for testing pest control efficiency, it's important to consider how the manuscript addresses key aspects of this assessment.
Overall, the manuscript presents valuable research on an important topic related to genetic engineering in maize. With some minor revisions and improvements, it could be a valuable contribution to the field.
There are minor grammatical errors and awkward phrasings throughout the manuscript that should be corrected.
- In the first sentence 44, you can improve clarity and conciseness:
- Original: "Maize is one of the primary staple food crops and plays a pivotal role in sustaining global food security."
- Suggested: "Maize is a primary staple food crop that plays a pivotal role in global food security."
- In the following sentence 45, consider rephrasing for clarity:
- Original: "In maize growing areas, many biotic stresses affect production capacity and safety..."
- Suggested: "In regions where maize is grown, various biotic stresses can impact both production capacity and safety..."
- Awkward phrasing in this sentence 52:
- Original: "Transgenic maize expressing Vip3A (Vegetative insecticidal protein 3) protein from Bt are widely used in GM crops to control lepidopteran pests..."
- Suggested: "Transgenic maize expressing the Vip3A (Vegetative insecticidal protein 3) protein from Bt is widely employed in GM crops for controlling lepidopteran pests..."
- Consider revising this sentence 54for clarity:
- Original: "Because of their high control efficacy, transgenic Bt crops have increased consistently since their initial commercialization in the USA during the 1990s[11]."
- Suggested: "Due to their high effectiveness in pest control, transgenic Bt crops have seen consistent growth since their initial commercialization in the USA during the 1990s[11]."
- Awkward phrasing in this sentence 56:
- Original: "Although Bt maize has been widely planted at multiple countries in recent decades, its biosafety is still controversial in certain countries [12,13]."
- Suggested: "While Bt maize has been extensively cultivated in numerous countries in recent decades, its biosafety remains a subject of controversy in certain regions [12,13]."
- Consider rephrasing this sentence 79for clarity:
- Original: "This promising Cre/loxP-mediated system has previously been applied in transgenic tobacco, rice and oilseed rape for manipulating transgenes expression in desired tissues 80 [26,27,29]."
- Suggested: "The promising Cre/loxP-mediated system has previously been employed in transgenic tobacco, rice, and oilseed rape to regulate the expression of target genes in specific tissues [26,27,29]."
The manuscript could benefit from a thorough proofreading to ensure that language issues do not detract from the clarity of the research.
Author Response
Dear Reviewer #1,
Thank you very much for your careful work and efforts with the many suggestions. We convinced that these valuable suggestions helped significantly to improve our text. We hope that the paper become better than the previous version was.
Comments to the Author
- The method described in the manuscript, which is based on the Cre/loxP-mediated system for manipulating gene expression in maize, is not inherently novel as the Cre/loxP system has been widely used in genetic engineering for many years. This system is a well-established and proven tool for controlling gene expression in various organisms, including plants. Therefore, the novelty of the research likely lies in the specific application of the Cre/loxP system to manipulate the expression of the Vip3A gene in maize, with a focus on confining its expression to specific tissues.
Response: Thank you for your valuable comments. We absolutely agreed with your opinion. Yes, the novelty of the research lies in application of the Cre/loxP system to manipulate the expression of the Vip3A gene in maize green tissues, with no accumulation in edible parts , which may be more acceptable to the public.
- In terms of references, it is essential in scientific research to provide a solid foundation of relevant literature to support the research idea and demonstrate awareness of existing work in the field. Upon reviewing the manuscript, it appears that the authors have included references to key studies and relevant research. They reference the use of the Cre/loxP system in other transgenic plants like tobacco, rice, and oilseed rape, which helps establish the context for their research.
However, it's always a good practice to ensure that the references cover a broad spectrum of related work, especially if there have been significant developments in the field that may impact the current research. Additionally, the authors could further strengthen their argument by discussing how their approach builds upon or differs from previous applications of the Cre/loxP system in plants.The inclusion of relevant references supports the research idea, but authors should ensure that they cover the most up-to-date and pertinent literature in their field of study.
Response: Special thanks for your valuable comments. These comments are great significance to guide our research on maize genetic engineering. To our knowledge, the Cre/loxP system in genetic engineering field has been applied in many transgenic plants. We had tried our best to collect the most relevant references about applications of the Cre/loxP system in plants for our current research. The typical differences between our approach and previous applications are that the tissue-specific expression and switch design of transgenes in plant genetic engineering. In most previous studies, the Cre/loxP system were usually developed to delete the introduced transgenes from the host genome, and the removed process may have potential risks for causing much damage and affecting multiple agronomic traits. In current research, the transgene Vip3A is always present in the green tissues and just shut down or turn on the expression without removing DNA from the plant genome, which could be little effect on agronomic traits and might be applied more widely. Following your suggestion, we have added more information to describe the differences between our approach and previous applications for GM plants in the section of discussion. (Please see new revised version).
- also, the provided excerpt from the manuscript mentions the assessment of pest control efficiency in hybrid maize but does not provide specific details regarding the experimental setup or the sample size. To evaluate the methodology for testing pest control efficiency, it's important to consider how the manuscript addresses key aspects of this assessment.
Response: Thank you for your valuable comments. We absolutely agreed with your opinion. For the bioactivity of transgenic maize against two pests in the laboratory and field, The each maize of the KEY × LOCK hybrids and controls were setup with three trays for a single larval infestation in the laboratory insect-bioassay. For field test, a total of 90 experimental hybrid transgenic maize was chosen randomly and investigated in current study. Following the suggestion, we have added more detailed information to describe the experimental design replicates in the section of materials and methods. (Please see new revised version).
- Overall, the manuscript presents valuable research on an important topic related to genetic engineering in maize. With some minor revisions and improvements, it could be a valuable contribution to the field.
Response: Special thanks for your kindly appraisements.
There are minor grammatical errors and awkward phrasings throughout the manuscript that should be corrected.
- In the first sentence 44, you can improve clarity and conciseness:
Original: "Maize is one of the primary staple food crops and plays a pivotal role in sustaining global food security."
Suggested: "Maize is a primary staple food crop that plays a pivotal role in global food security."
Response: Special thanks for your careful review. Following the comment, corrected as suggested. (Please see new revised version).
- In the following sentence 45, consider rephrasing for clarity:
Original: "In maize growing areas, many biotic stresses affect production capacity and safety..."
Suggested: "In regions where maize is grown, various biotic stresses can impact both production capacity and safety..."
Response: Special thanks for your careful review. Following the comment, corrected as suggested. (Please see new revised version).
- Awkward phrasing in this sentence 52:
Original: "Transgenic maize expressing Vip3A (Vegetative insecticidal protein 3) protein from Bt are widely used in GM crops to control lepidopteran pests..."
Suggested: "Transgenic maize expressing the Vip3A (Vegetative insecticidal protein 3) protein from Bt is widely employed in GM crops for controlling lepidopteran pests..."
Response: Special thanks for your careful review. Following the comment, corrected as suggested. (Please see new revised version).
- Consider revising this sentence 54for clarity:
Original: "Because of their high control efficacy, transgenic Bt crops have increased consistently since their initial commercialization in the USA during the 1990s[11]."
Suggested: "Due to their high effectiveness in pest control, transgenic Bt crops have seen consistent growth since their initial commercialization in the USA during the 1990s[11]."
Response: Special thanks for your careful review. Following the comment, corrected as suggested. (Please see new revised version).
- Awkward phrasing in this sentence 56:
Original: "Although Bt maize has been widely planted at multiple countries in recent decades, its biosafety is still controversial in certain countries [12,13]."
Suggested: "While Bt maize has been extensively cultivated in numerous countries in recent decades, its biosafety remains a subject of controversy in certain regions [12,13]."
Response: Special thanks for your careful review. Following the comment, corrected as suggested. (Please see new revised version).
- Consider rephrasing this sentence 79 for clarity:
Original: "This promising Cre/loxP-mediated system has previously been applied in transgenic tobacco, rice and oilseed rape for manipulating transgenes expression in desired tissues 80 [26,27,29]."
Suggested: "The promising Cre/loxP-mediated system has previously been employed in transgenic tobacco, rice, and oilseed rape to regulate the expression of target genes in specific tissues [26,27,29]."
Response: Special thanks for your careful review. Following the comment, corrected as suggested. (Please see new revised version).
- The manuscript could benefit from a thorough proofreading to ensure that language issues do not detract from the clarity of the research.
Response: Special thanks for your careful review. We are sorry for our unidiomatic description in the original draft. Before sending the paper to the journal Insects, the manuscript has been polished by an English Editing company to improve the language. We made the improvements accordingly. For this concern,, we tried our best and went through the text again to make it more better. (Please see new revised version).
Reviewer 2 Report
The authors report on the use of the Cre/loxP system to preferentially express Vip3A in certain tissues. This is a MS resubmitted to a different journal. The authors are requested to address the following suggestions
- The explants that were used for plant transformation should be added to the plant material and transformation session.
- Copy numbers of each transgene in the parents that were used for the crossing and hybrid plants used for expression analyses should be presented.
- Sequencing data (or Southern blots) should be provided to demonstrate that removal of the terminator occurred only in green tissues but not the other tissues.
- Developmental stages of the plant tissues that were used for qRT-PCR and ELISA should be stated.
- A brief description of the gene suppression MOA by the NosT inserted between the promoter and coding region would be helpful.
6. Using a stronger promoter to drive Vip3A expression would lead to higher levels of the Vip3A protein in target tissues to get acceptable insect control efficacy.
English is okay but could use some improvement
Author Response
Dear Reviewer #2,
Thank you very much for your careful work and efforts with the many suggestions. We convinced that these valuable suggestions helped significantly to improve our text. We hope that the paper become better than the previous version was.
Comments and Suggestions for Authors
The authors report on the use of the Cre/loxP system to preferentially express Vip3A in certain tissues. This is a MS resubmitted to a different journal. The authors are requested to address the following suggestions
- The explants that were used for plant transformation should be added to the plant material and transformation session.
Response: Thank you for your valuable comments. Following the comment, added as suggested. (Please see new revised version).
- Copy numbers of each transgene in the parents that were used for the crossing and hybrid plants used for expression analyses should be presented.
Response: Thank you for your valuable comments. Although we have been trying to check copy numbers of the transgenes in our transgenic maize by using Southern blots, however, we failed to obtain positive results duo to technical reasons. You know, the Southern experiment is always affected by many factors that make it difficult to achieve good results. We have been trying the experiment for many times, but no bands appearing. In the end, we had to give up it. For this concern, we have no related results about the gene copy, sorry for that.
- Sequencing data (or Southern blots) should be provided to demonstrate that removal of the terminator occurred only in green tissues but not the other tissues.
Response: Thank you for your valuable comments. As we response in the above, we failed to obtain positive results duo to technical reasons. So, we have no more evidences to demonstrate that removal of the terminator occurred only in green tissues, sorry for that. We can detect the relative expression levels and concentration of the Vip3A in the various tissues to determine the excision reaction occurred in target green tissues.
- Developmental stages of the plant tissues that were used for qRT-PCR and ELISA should be stated.
Response: Thank you for your valuable comments. Actually, Vip3A levels in the various tissues of LOCK and KEY × LOCK hybrids were detected at seedling, tillering, filling and mature stages during the trangenic maize growth period in our research. The model of tissue-specific expression and the content pattern of Vip3A in the various tissues of LOCK plants and KEY × LOCK hybrids were similar at different growth stages. So, we preferred to select the typical mature stage to demonstrate the final Vip3A levels in current study.
- A brief description of the gene suppression MOA by the NosT inserted between the promoter and coding region would be helpful.
Response: Thank you for your valuable comments. We absolutely agreed with your opinion. Following the suggestion, we have added more detailed information to describe the design of the Cre/loxP-mediated approach in the revised version. (Please see new revised version).
- Using a stronger promoter to drive Vip3A expression would lead to higher levels of the Vip3A protein in target tissues to get acceptable insect control efficacy.
Response: Thank you for your valuable comments. You are quite right. For our Cre/loxP appoach, the green tissue-specific promoter Zm1rbcS with moderate drive capability was chosen to induce the expression of the simple gene Cre in KEY plants. On the other side, the big gene Vip3A must be driven by a powerful promoter to produce a sufficient quantity of insecticidal protein.
- Comments on the Quality of English Language
English is okay but could use some improvement
Response: Special thanks for your kindly appraisements. We made the improvements accordingly.
Round 2
Reviewer 2 Report
This manuscript is a resubmission detailing the engineering of expression of Vip3A in Green Tissues as a Feasible Approach for controlling Asian Stem Borer and Fall Armyworm in Maize. The authors have responded to this reviewer's comments and unfortunately, were able only to comply with several of this reviewer's suggestions.
As suggested previously, the final expression and toxicity against the target pests is only moderate and probably not commercial (and this is with Vip3A, probably the most toxic insecticidal protein against FAW; would this current system be appropriate with less toxic proteins?). Therefore, this reviewer wonders whether this approach is feasible to achieve commercial level efficacy (and with less toxic proteins), and in doing so, what will be the effect on non-green tissues? The authors need to address this in their discussion and elaborate on statements that their work is essentially proof of concept.
Several specific comments:
Table 2: Change second O. furnacalis to S. frugiperda in both Larval Mortality and Plant leaf damage scale.
Lines 347-348: Please remove the term "pyramided" (unless authors can justify) as the authors are expressing only Vip3A and not another mode of action against these pests
Author Response
Dear Reviewer,
Thank you very much for your careful work and efforts on our article. We have studied these valuable suggestions carefully and made the improvements accordingly.
Comments to the Author
This manuscript is a resubmission detailing the engineering of expression of Vip3A in Green Tissues as a Feasible Approach for controlling Asian Stem Borer and Fall Armyworm in Maize. The authors have responded to this reviewer's comments and unfortunately, were able only to comply with several of this reviewer's suggestions.
As suggested previously, the final expression and toxicity against the target pests is only moderate and probably not commercial (and this is with Vip3A, probably the most toxic insecticidal protein against FAW; would this current system be appropriate with less toxic proteins?). Therefore, this reviewer wonders whether this approach is feasible to achieve commercial level efficacy (and with less toxic proteins), and in doing so, what will be the effect on non-green tissues? The authors need to address this in their discussion and elaborate on statements that their work is essentially proof of concept.
Response: Special thanks for your valuable comments. These comments are great significance to guide our research on maize genetic engineering. We absolutely agreed with your opinion that the toxicity of the Vip3A against the target pests is only moderate and probably not reaching to commercial level. The current research was a preliminary step in manipulating Vip3A expression in maize green tissues. Further work including promoting the effectiveness of the tissue-specific promoter or excision reaction efficiency of this approach, should be explored and evaluated deeply. In brief, the novelty of the current research lies in application of the Cre/loxP system to manipulate the expression of the Vip3A gene in maize green tissues, with no accumulation in edible parts, which may be more acceptable to the public.
As mentioned in the text, the presence of Cre protein in green tissues is necessary for the effective control of Vip3A protein in defined tissues. At present stage, the moderate toxicity of Vip3A protein is probably depend on the efficiency of the excision reaction. In the approach, Cre recombinase from KEY plants, responsible for specific excision of the blocked element NosT in LOCK plants, may not be completely removed the NosT resulting in moderate level toxic protein in green tissues. To further optimize this approach, the green tissue-specific promoter Zm1rbcS can be replaced with a better promoter to increase the efficiency of the excision reaction, which maybe improve the Vip3A protein to achieve commercial level efficacy. On the other hand, the toxic level of Vip3A protein was just an average of different cross combinations including low or high level insect-resistance. The most cross combinations that reach commercial level efficacy should be selected for further breeding program. Given that, the Cre/loxP system could be recommended to potentially apply in GM maize and may achieve commercial level efficacy in next step. For this concern, we have added more detailed information to describe the concern in the discussion section. (Please see new revised version).
Several specific comments:
Table 2: Change second O. furnacalis to S. frugiperda in both Larval Mortality and Plant leaf damage scale.
Response: Special thanks for your careful review. Following the comment, corrected as suggested. (Please see new revised version).
Lines 347-348: Please remove the term "pyramided" (unless authors can justify) as the authors are expressing only Vip3A and not another mode of action against these pests.
Response: Special thanks for your careful review. Following the comment, corrected as suggested. (Please see new revised version).